# Conservation of Importin α Function in Apicomplexans: Ivermectin and GW5074 Target *Plasmodium falciparum* Importin α and Inhibit Parasite Growth in Culture

**DOI:** 10.3390/ijms232213899

**Published:** 2022-11-11

**Authors:** Sujata B. Walunj, Chunxiao Wang, Kylie M. Wagstaff, Swati Patankar, David A. Jans

**Affiliations:** 1Molecular Parasitology Lab., Department of Biosciences and Bioengineering, Indian Institute of Technology Bombay, Powai, Mumbai 400076, India; 2Nuclear Signalling Lab., Department of Biochemistry and Molecular Biology, Monash Biomedicine Discovery Institute, Monash University, Monash, VIC 3800, Australia

**Keywords:** nuclear import, importins, malaria, toxoplasmosis, nuclear import inhibitors

## Abstract

Signal-dependent transport into and out of the nucleus mediated by members of the importin (IMP) superfamily of nuclear transporters is critical to the eukaryotic function and a point of therapeutic intervention with the potential to limit disease progression and pathogenic outcomes. Although the apicomplexan parasites *Plasmodium falciparum* and *Toxoplasma gondii* both retain unique IMPα genes that are essential, a detailed analysis of their properties has not been performed. As a first step to validate apicomplexan IMPα as a target, we set out to compare the properties of *P. falciparum* and *T. gondii* IMPα (PfIMPα and TgIMPα, respectively) to those of mammalian IMPα, as exemplified by *Mus musculus* IMPα (MmIMPα). Close similarities were evident, with all three showing high-affinity binding to modular nuclear localisation signals (NLSs) from apicomplexans as well as Simian virus SV40 large tumour antigen (T-ag). PfIMPα and TgIMPα were also capable of binding to mammalian IMPβ1 (MmIMPβ1) with high affinity; strikingly, NLS binding by PfIMPα and TgIMPα could be inhibited by the mammalian IMPα targeting small molecules ivermectin and GW5074 through direct binding to PfIMPα and TgIMPα to perturb the α-helical structure. Importantly, GW5074 could be shown for the first time to resemble ivermectin in being able to limit growth of *P. falciparum*. The results confirm apicomplexan IMPα as a viable target for the development of therapeutics, with agents targeting it worthy of further consideration as an antimalarial.

## 1. Introduction

Apicomplexans are single-celled, obligate intracellular eukaryotes, and include *Plasmodium* species and *Toxoplasma gondii* that are able to cause severe disease in humans [1,2]. Malaria caused by *Plasmodium* resulted in close to 240 million cases and >600,000 deaths (predominantly children under the age of five) in 2020 alone [3]. By comparison, *T. gondii* chronically infects about one-third of the human population worldwide; although c. 80% of infections are asymptomatic, infection of pregnant women can lead to severe outcomes in newborn babies (e.g., hydrocephaly, microcephaly), while immunocompromised individuals, such as those with HIV/AIDS, can develop severe toxoplasmosis (e.g., encephalitis, ocular toxoplasmosis) [4,5].

The only vaccine thus far approved is RTS, S/AS01(RTS, S) for children living in malarial endemic areas [6]; WHO believes it will reduce severe malarial disease by 30% in vaccinated children, but with the caveat that malarial transmission will not be reduced significantly, and, hence, malarial endemicity will likely be reinforced (see [6]). Artemisinin Combination Therapy (ACT) is currently the antimalarial treatment of choice (World Malaria Report, 2018), even though resistance to artemisinin and the other agents used in ACT has recently emerged in Southeast Asia [7,8]. In the case of toxoplasmosis, there are no currently approved vaccines; the first-line therapy to treat acute cases of toxoplasmosis is the combination of pyrimethamine (PYR) and sulfadiazine (SDZ), but such treatment is of poor tolerance and there are rare fatal side-effects, and even drug resistance has been reported [9]. Overall, there is clearly a need to renew efforts to identify new drugs to limit malaria and toxoplasmosis.

Trafficking into and out of the nucleus mediated by members of the importin (IMP) superfamily of α and β proteins is of increasing interest as a drug target for a variety of human diseases, including viral infections [10,11,12,13]. The best understood nuclear import pathway involves nuclear localisation signal (NLS) recognition by IMPα in a heterodimer with IMPβ1 to which it binds through its IMPβ1-binding (IBB) domain [14,15,16]; the trimeric IMPα/β-NLS-containing protein complex then docks at the nuclear pore complex (NPC) through IMPβ1, and the complex dissociates inside the nucleus upon binding of the monomeric guanine nucleotide-binding protein Ran in activated GTP-bound form [17]. Dysregulation of nucleocytoplasmic transport can have an impact on a range of cellular processes, including cell signaling, proliferation, growth, and differentiation [10,18,19], with inhibition of nuclear transport holding great potential for therapeutic intervention [10,11,13]. High throughput screening (HTS) has been used to identify small molecule inhibitors of IMPα-dependent nuclear import, such as GW5074 and ivermectin, that limit infection by human pathogenic viruses, such as human immunodeficiency virus (HIV-1), dengue, Zika, West Nile virus, and SARS-CoV2 [20,21,22,23,24,25].

Although *P. falciparum* and *T. gondii* both retain unique IMPα genes that are essential [26,27], detailed functional or biophysical analysis has not been performed beyond initial studies into IMPα from *P. falciparum* (PfIMPα) [28], meaning that the extent to which apicomplexan IMPα resembles mammalian IMPα, and the potential effect thereon of inhibitors, such as ivermectin, is unknown. Here we set out to begin to address this question, by comparing the properties of PfIMPα and *T. gondii* IMPα (TgIMPα) to those of mammalian IMPα, as exemplified by *Mus musculus* IMPα (MmIMPα), for the first time. Close similarities were evident, with all three showing high-affinity binding to NLSs from apicomplexans as well as Simian virus SV40 large tumour antigen (T-ag); PfIMPα and TgIMPα were also capable of binding to mammalian IMPβ1 (MmIMPβ1) with high affinity. Strikingly, NLS binding by PfIMPα and TgIMPα could be inhibited by ivermectin and GW5074, through binding directly to PfIMPα and TgIMPα to perturb structure. Importantly, we could show for the first time that GW5074 resembled ivermectin in being able to limit the growth of *P. falciparum*. The conservation of the properties of IMPα from mammals to apicomplexans implies that agents such as ivermectin and GW5074 are worthy of further consideration as therapeutics for malaria and, potentially, other parasitic infections.

## 2. Results

### 2.1. Apicomplexan NLSs Can Be Recognised with High Affinity by Mammalian IMPα/β1; Apicomplexan IMPαs Can Bind a Heterologous NLS with High Affinity

We initially set out to compare the NLS-binding ability of recombinantly expressed PfIMPα and TgIMPα with that of the mammalian IMPα/β1 heterodimer from *M. musculus* (MmIMPα/β1) using an established AlphaScreen binding assay, as previously conducted [20,24]; the assay included apicomplexan NLSs and the well-characterised NLS from simian virus SV40 large tumour antigen (T-ag) in the form of NLS-GFP fusion proteins expressed in bacteria. *P. falciparum* trimethylguanosine synthase (TGS1) is an RNA methylase involved in the hypermethylation of 5′ ends of non-coding RNAs, with residues 226–261, containing a putative bipartite NLS and able to target GFP to the nucleus in a heterologous system [29]. *T. gondii* histone acetyltransferase GCN5 is a nuclear protein involved in gene regulation, with residues 94–99 (RKRVKR, single-letter amino acid code) shown to be sufficient and essential to target GFP to the nucleus in *T. gondii* tachyzoites [30].

As previously, MmIMPα/β1 bound to the T-ag-NLS with high affinity (apparent dissociation constant, Kd, of c. 3 nM); strikingly, it also recognised the apicomplexan NLSs with comparably high binding affinities (Figure 1 right; see Table 1 for pooled data). In similar fashion, PfIMPα and TgIMPα both bound their respective NLSs (TGS1-NLS and GCN5-NLS, respectively) with high (low nM) binding affinity (Figure 1 bottom row; Table 1); strikingly again, both PfIMPα and TgIMPα showed high-affinity binding to the T-ag-NLS (Kd of 1–3 nM; Figure 1 top; Table 1). The clear implication is that NLS sequences from mammalian and apicomplexan systems may be largely comparable and interchangeable (see also [29]) and that apicomplexan IMPαs have conserved NLS-binding functionality, comparable to that of MmIMPα.

### 2.2. Apicomplexan IMPα Can Bind to MmIMPβ1 with High Affinity 

The results in Figure 1/Table 1 indicate that both apicomplexan IMPαs show high binding affinity in the absence of IMPβ1; this is consistent with the idea that, as previously shown for PfIMPα [28], apicomplexan IMPα appears to differ from mammalian IMPαs in showing reduced autoinhibition in the absence of IMPβ1. In the case of both *P. falciparum* and *T. gondii*, although IMPβ-like sequences have been identified in their respective genomes [31], a clear homolog of mammalian IMPβ1 has not yet been identified. To assess the potential ability of apicomplexan IMPα to heterodimerise with IMPβ1 for the first time, we tested PfIMPα and TgIMPα binding to *M. musculus* IMPβ1 (MmIMPβ1) using the AlphaScreen-based binding assay [24,32]. His-tagged PfIMPα and TgIMPα were titrated with increasing concentrations of biotinylated GST-MmIMPβ1, as shown in Figure 2 (left; see Table 2 for pooled data). His-tagged MmIMPα was tested in parallel for comparison (Figure 2 right panel), and the Kd for interaction between MmIMPα and MmIMPβ1 was in the low nM range (2.5 nM), as previously shown [24]. Interestingly, both PfIMPα and TgIMPα also showed high-affinity binding to MmIMPβ1 (Kd of 1.2–1.3 nM) (Figure 2; Table 2). Clearly, the results indicate that apicomplexan IMPα is sufficiently homologous to mammalian IMPαs to be able to heterodimerise with mammalian IMPβ1, testifying to the conservation of function.

### 2.3. Inhibitors of Mammalian IMPα Can Block Apicomplexan IMPα Interaction with NLSs and MmIMPβ1

Inhibitors of mammalian IMPα function are of key interest as anti-infectious agents [10,11,13]. One of the first described is the small molecule macrocyclic lactone ivermectin, produced by the bacterium *Streptomyces avermitilis*, that is approved by the US Food and Drug Administration for parasitic infections. Ivermectin has also been shown to be a potent antiviral agent [11,13,25,33] through its ability to bind within the armadillo (ARM) repeat domain of mammalian IMPα to induce structural changes that prevent NLS recognition and binding to IMPβ1 to form the IMPα/β1 complex; it can also dissociate preformed IMPα/β complexes [32]. Chemically distinct from ivermectin, GW5074 (3-(3,5-dibromo-4-hydroxybenzyliden)-5-iodo-1,3-dihydroindol-2-one) binds to mammalian IMPα with analogous structural effects on IMPα and inhibitory effects on NLS recognition/IMPβ1 binding by IMPα [24]. GW5074 has not previously been analysed for effects in apicomplexan systems, but ivermectin has been shown to block nuclear transport of PfSRP polypeptides into the nucleus in *P. falciparum* parasites [34], although the mechanistic details have not been examined. We set out to test the activity of these inhibitors with respect to NLS binding by PfIMPα and TgIMPα, with NLS binding by MmIMPα analysed in parallel (Figure 3; Table 3). As expected, ivermectin and GW5074 inhibited MmΔIBBIMPα-SV40 T-ag with IC_50_ values of 5–6 µM, consistent with published data [24,32]. Strikingly, both compounds inhibited NLS-recognition by Pf and TgIMPα at low µM concentrations, with IC_50_ values of 5.0 µM for ivermectin and 6–8 µM for GW5074 (Figure 3, Table 3). The results clearly indicate that chemical inhibitors known to target mammalian IMPα can inhibit apicomplexan IMPα, underlining the structural and functional conservation of IMPα from apicomplexans to mammalian systems.

As indicated, both ivermectin and GW5074 have been previously shown to inhibit interaction between mammalian IMPα and IMPβ1 [24,32]. We tested their activity towards apicomplexan IMPα (Figure 4; Table 4). As previously observed, binding of MmIMPα:MmIMPβ1 was inhibited strongly by both agents, with IC_50_ values of c. 3 and 22 µM (Figure 4, Table 4); strikingly, ivermectin also inhibited binding of PfIMPα and TgIMPα to MmIMPβ1 at low µM concentration (IC_50_ of 2–7 µM), with comparable results for GW5074 (IC_50_ values of 10–40 µM). Clearly, ivermectin and GW5074 are able to act on apicomplexan IMPα in analogous fashion to MmIMPα in terms of the ability to bind to IMPβ1, the results further underlining the structural and functional conservation of IMPα from apicomplexans to mammalian systems.

### 2.4. Ivermectin and GW5074 Appear to Bind Directly to Apicomplexan IMPαs to Have an Impact on Conformation

Previous analysis for MmIMPα using several biophysical approaches had established that the mechanism by which both ivermectin and GW5074 have an impact on NLS and IMPβ1 binding by MmIMPα is by direct binding to the ARM repeat domain of IMPα to perturb structural conformation [24,32]. Here, we employed far-UV CD spectroscopy, as previously [24], to confirm these results for MmIMPα and expand analysis to PfIMPα and TgIMPα. The CD spectra of PfIMPα, TgIMPα, and MmIMPα all showed a double minimum at 208 and 222 nm (Figure 5a), consistent with a predominantly α-helical structure [24,35]. Quantitative estimation of the % α-helicity (see Section 4.3) revealed that the apicomplexan IMPαs were c. 63 % α-helical, compared to c. 70 % for MmIMPα (Figure 5b).

CD spectra were also analysed in the presence of increasing concentrations of ivermectin and GW5074, with both agents having a marked impact on the structure, especially at high concentrations (Figure 5a,b), consistent with the idea that both bind to the proteins directly to perturb structure. PfIMPα appeared to be particularly susceptible to both agents, with 30 µM GW5074 reducing α-helicity by more than a half (Figure 5a,b) to a residual value of 30%, which fell below 10% in the presence of 80 µM GW5074. Ivermectin had comparable effects, reducing α-helicity by two-thirds to a residual value of c. 20% at 80 µM. The results, overall, indicate that both ivermectin and GW5074 can bind to and destabilise the structure of both PfIMPα and TgIMPα in analogous fashion to their known effects on mammalian IMPα, further underlining the conservation of structure and function between apicomplexan and mammalian IMPαs.

**Figure 5 ijms-23-13899-f005:**
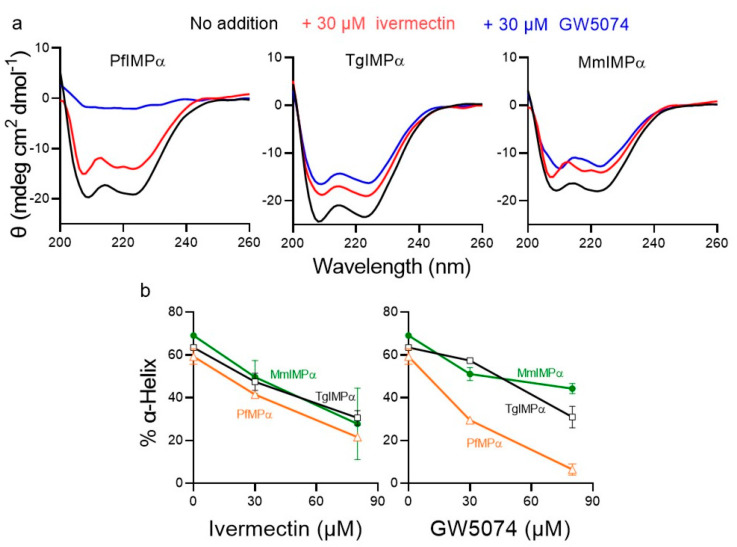
CD spectra for apicomplexan and mammalian IMPαs in the absence and presence of ivermectin and GW5074. CD spectra were collected for PfIMPα, TgIMPα, and MmIMPα in the absence or presence of 30 μM and 80 μM ivermectin and GW5074. (**a**) Spectra shown are from a single experiment, representative of two independent experiments for 30 μM concentration of ivermectin and GW5074. Note: θ is ellipticity. (**b**) The α-helical content of the respective IMPαs was estimated as previously (see Section 4.3) from spectra as per (**a**). Results represent the mean ± SD for two independent experiments.

### 2.5. Ivermectin and GW5074 Can Limit the Proliferation of P. falciparum In Vitro

Ivermectin has previously been shown to possess robust antimalarial activity [36,37], the study here providing the mechanistic basis for this through its action on PfIMPα as documented above, but GW5074 has not been previously tested against *P. falciparum* parasites. To test the potential antimalarial action of GW5074, we compared the activity of GW5074 to ivermectin against the asexual stages of *P. falciparum* parasites, using an established HRP2-based (histidine-rich protein 2) ELISA [38,39,40]. The assay uses a sandwich ELISA to measure HRP2 levels as an indicator of parasite growth; the clinically prescribed drug dihydroartemisinin (DHA) was used as a positive control.

DHA showed potent activity in this assay, with an IC_50_ value of c. 1 nM, with the value for ivermectin c. 1 µM (Figure 6, Table 5). Importantly, the IC_50_ value determined for GW5074 was c. 6 µM (Figure 6, Table 5), confirming its antimalarial activity. That agents targeting PfIMPα, such as GW5074 and ivermectin, have robust anti-parasitic activity confirms PfIMPα/nuclear transport in *P. falciparum* as a viable target to develop antimalarial therapeutics.

## 3. Discussion

This study is the first to document the close structural and functional conservation between apicomplexan and mammalian IMPα, and, in so doing, confirm apicomplexan IMPα as a viable target for the development of antimalarials. Prior to the present study, relatively little was known regarding the apicomplexan nuclear transport system, with bioinformatics analysis suggesting some conservation of some of the basic transport components [31]. The analysis here, firstly, confirms that NLSs functional in mammalian and apicomplexan systems can be recognised by apicomplexan and mammalian IMPαs respectively, with nM affinity; thus, NLSs in apicomplexan and mammalian systems are essentially interchangeable (see also [29]). Further, even though putative Pf and TgIMPβ1 proteins show only 20–30% sequence similarity to MmIMPβ1, both Pf and TgIMPα can interact with MmIMPβ1 with nM affinity, in comparable fashion to MmIMPα; the clear implication here is that function has been conserved through evolution, despite the low-sequence homology [41,42,43]. Finally, this study establishes for the first time that NLS and IMPβ1 binding by apicomplexan IMPα can be inhibited by ivermectin and GW5074, which are both agents that bind to mammalian IMPα to disrupt structure; CD analysis indicates that Pf and TgIMPα are very similar to mammalian IMPα in being largely α-helical, presumably, key to making them susceptible to ivermectin/GW5074 binding in the same way.

Importantly, the study here confirms apicomplexan IMPα as a viable therapeutic target, with both ivermectin and GW5074 confirmed here to have robust antimalarial activity (see Figure 6). This is the first time GW5074 has been shown to have antimalarial activity, and, although ivermectin has previously been shown to possess antiparasitic activity [36,37], this study is the first to implicate binding to PfIMPα/disruption of its role in nuclear import as the likely mechanism of the action of ivermectin in this context (see [32,44]). That structurally unrelated compounds such as GW5074 and ivermectin are able to target PfIMPa and both have antimalarial action supports this idea (see also [45]), but, of course, it should be remembered that both compounds are known to have other activities in cells, e.g., ivermectin is known to target the glutamate-gated chloride channels of helminths [46,47,48] and has even been reported to inhibit RNA helicase activity in the context of DENV [49]. Clearly, although beyond the scope of the present study, confirmation of the inhibitory action of ivermectin and GW5074 in the context of the malarial parasite is essential to demonstrate formally that these agents do, indeed, block nuclear import in apicomplexans. Together with our recent study identifying a number of agents able to inhibit NLS binding by PfIMPα through analogous mechanisms of structural perturbation [45], the study here supports the idea that apicomplexan IMPα is an exciting therapeutic target for the future. Understanding the precise binding site on the respective apicomplexan and mammalian IMPα may well be the key to developing agents that specifically target apicomplexan and not mammalian host IMPα, and, thereby, the key to a new class of antimalarials that are selective and efficacious.

Although GW5074 is not approved for human clinical use, ivermectin has been used worldwide for more than 40 years to combat/prevent parasitic infections [46,47,48]. Its ability to partition to particular tissues and remain stable there for months implies that ivermectin is worth considering further in the immediate future not only as a broad-spectrum antiviral [11,32,33,47,50,51], through its effects on mammalian IMPα, but also as an antimalarial [52,53] that can target PfIMPα. This exciting possibility is the focus of further work in our respective laboratories.

## 4. Materials and Methods

### 4.1. Protein Expression, Purification, and Use in AlphaScreen Assay

PfIMPα, TgIMPα, MmIMPα, ΔIBBMmIMPα, and MmIMPβ1 GST fusion proteins were expressed and purified essentially as previously [20,21,22,24,32,45]. His-tagged T-ag-NLS-GFP, TGS1-NLS-GFP, GCN5-NLS-GFP, PfIMPα, and TgIMPα proteins were all purified using Ni^2+^-affinity chromatography, as previously [24,32,45]. Biotinylation of GST-tagged proteins was carried out using the Sulfo-NHS-Biotin reagent (Pierce, Rockford, IL, USA), as described previously [20]. The AlphaScreen binding assay was performed as previously [20,21,22,24,32,45].

### 4.2. Inhibitors

GW5074 and ivermectin were sourced from Sigma-Aldrich, St. Louis, MO, USA, and dissolved in dimethyl sulfoxide (DMSO) as 10 mM stock solutions.

### 4.3. CD Spectroscopy

CD spectroscopy was used to study the binding of ivermectin and GW5074 to PfIMPα, TgIMPα and MmIMPα, as previously described [24,32,45]. CD spectra of the proteins at 0.1 mg/mL concentration were recorded from 200–260 nm using Jasco CD spectrometer (Jasco, Portland, OR, USA). CD spectra of the IMPαs were also measured in the presence of 30 µM and 80 µM concentrations of the ivermectin and GW5074 compounds. Percentage α-helix content was calculated from the ellipticity at 222 nm as described in [51]. CD Multivariate secondary structure estimation (SSE) analysis program was used to estimate the percentage of α-helix content [24,32,54].

### 4.4. P. falciparum Culture and Growth Inhibition Assay

*P. falciparum* 3D7 strain was cultured and maintained in human red blood cells (RBCs) using RPMI (Roswell Park Memorial Institute) 1640 medium supplemented with 0.5% Albumax (GibcoTM, Waltham, MA, USA), 50 mg/L hypoxanthine (Sigma-Aldrich, St. Louis, MO, USA), 2 g/L D-glucose (Sigma-Aldrich, St. Louis, MO, USA), 2 g/L sodium bicarbonate (Sigma-Aldrich, St. Louis, MO, USA), and 56 mg/L of gentamicin (Abbott, Chicago, IL, USA) at 37 °C in 5% CO_2_ in a humidified incubator, according to the standard procedures [55]. Cultures presynchronised using 5% D-sorbitol (Sigma-Aldrich, St. Louis, MO, USA) to obtain predominantly the ring stage of the *P. falciparum* life cycle were used for the growth assays, as previously [56]. DHA (a gift from IPCA Laboratories, Mumbai, India) was used as a control. Growth assays in the absence and presence of inhibitors were performed using the HRP2 sandwich horseradish peroxidase-linked immunosorbent assay that measures HRP2 levels as an indicator of growth [39,40].

## Figures and Tables

**Figure 1 ijms-23-13899-f001:**
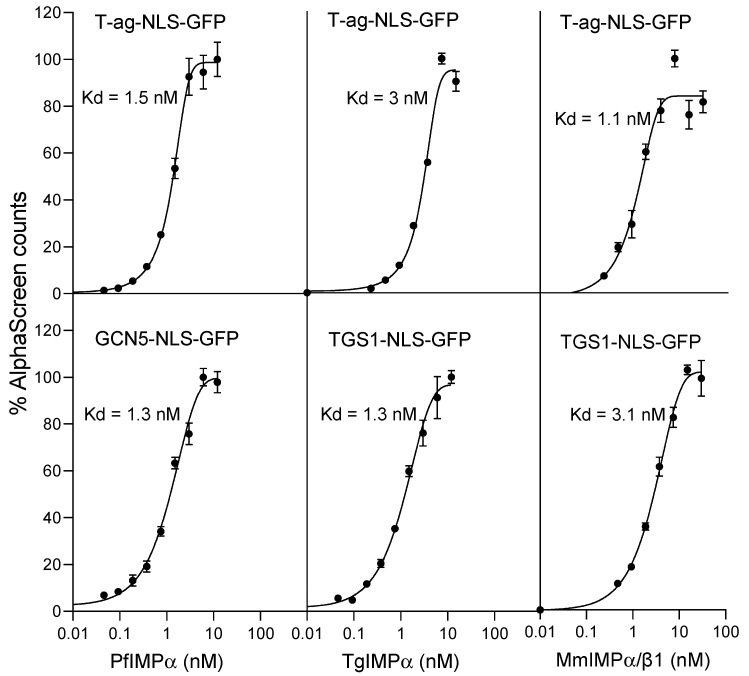
PfIMPα, TgIMPα, and MmIMPα/β1 all show high-affinity binding to NLSs from diverse sources. AlphaScreen technology was used to determine the binding affinity of biotinylated IMPαs binding to His-tagged NLS-GFPs (30 nM). Data points represent the mean ± SEM from quadruplet wells from a single typical experiment from a series of two independent experiments (see Table 1 for pooled data).

**Figure 2 ijms-23-13899-f002:**
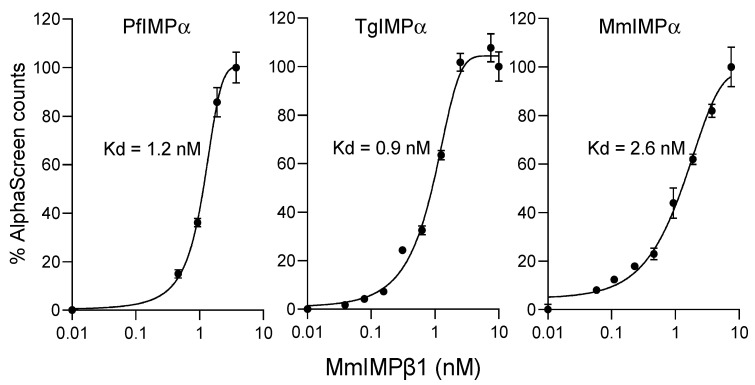
PfIMPα and TgIMPα can bind with high affinity to mammalian IMPβ1 in comparable fashion to MmIMPα. AlphaScreen technology was used to determine the Kd value of His-tagged IMPαs (30 nM) binding to biotinylated-GST-MmIMPβ1 (5 nM). Data points in the figures represent the mean ± SEM from quadruplet wells from a single typical experiment from a series of three independent experiments (see Table 2 for pooled data).

**Figure 3 ijms-23-13899-f003:**
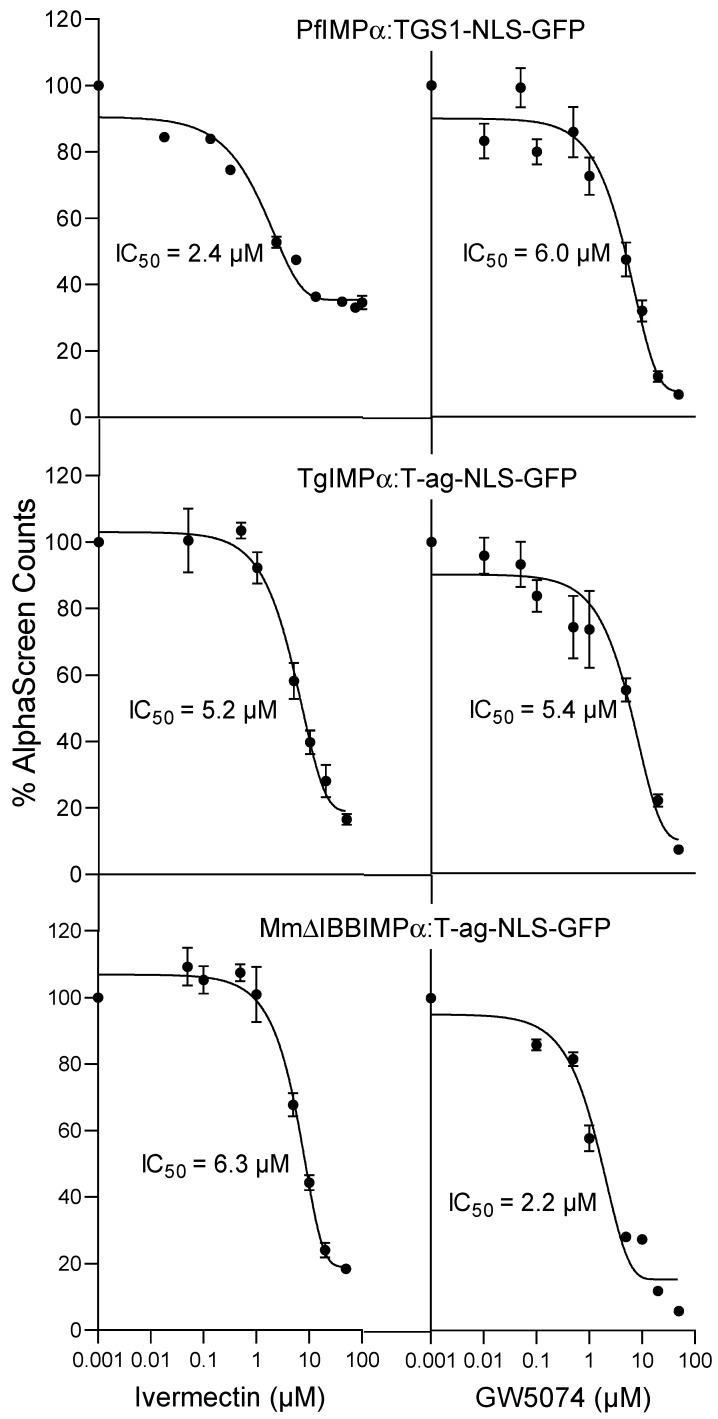
Ivermectin and GW5074 inhibit IMPα-NLS interaction at low micromolar concentration. AlphaScreen technology was used to determine the IC_50_ for inhibition by ivermectin and GW5074 binding of IMPαs (5 nM) to NLS (30 nM). Data represent the mean ± SEM for quadruplet wells from a single experiment, from a series of 3 independent experiments (see Table 3 for pooled data).

**Figure 4 ijms-23-13899-f004:**
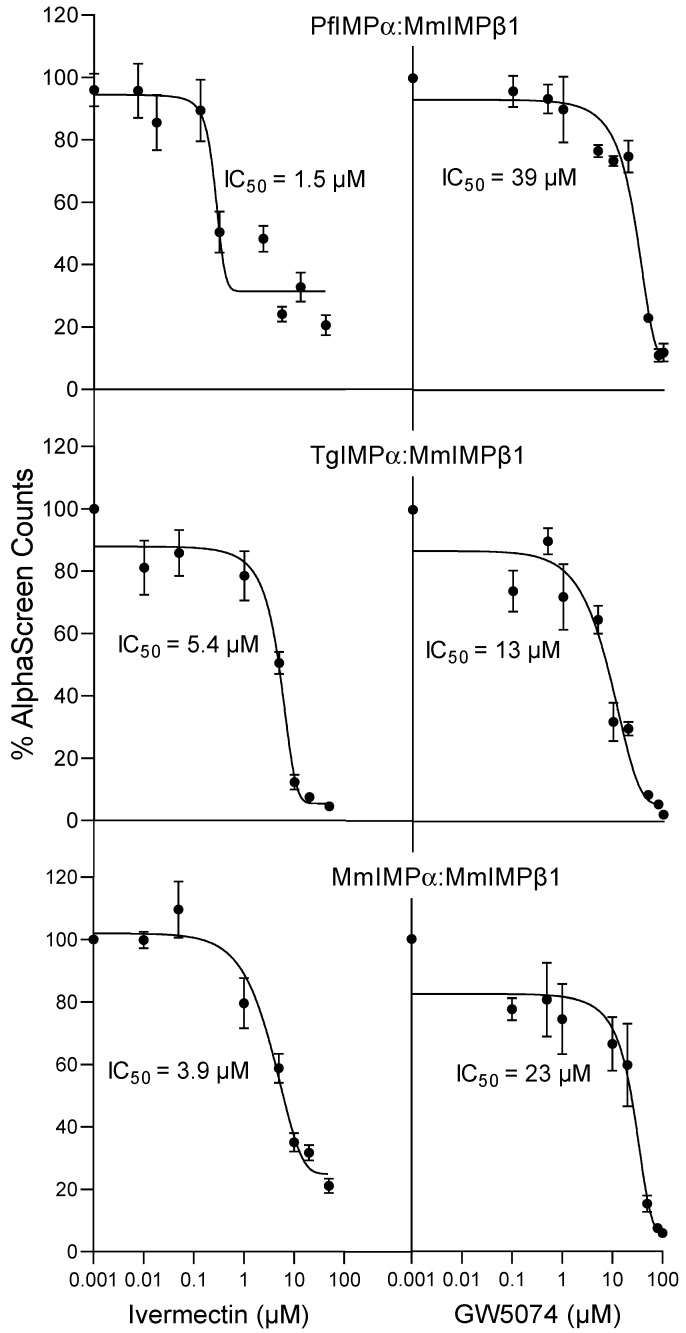
Ivermectin and GW5074 inhibit IMPα–IMPβ interaction. AlphaScreen technology was used to determine the IC_50_ for inhibition by ivermectin and GW5074-binding of IMPβ (30 nM) to IMPαs (30 nM). Data represent the mean ± SEM for quadruplet wells from a single experiment, from a series of three independent experiments (see Table 4 for pooled data).

**Figure 6 ijms-23-13899-f006:**
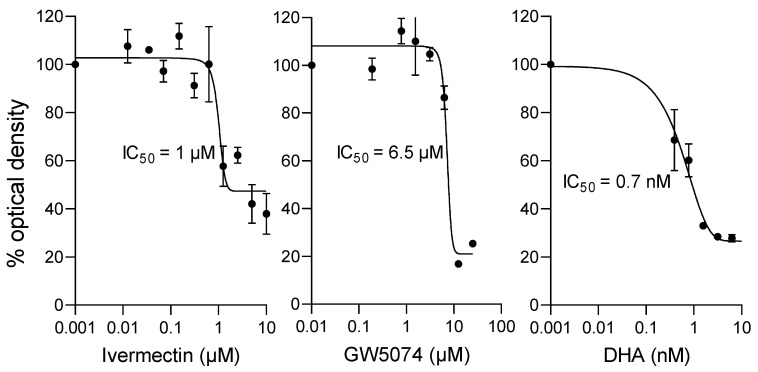
Ivermectin and GW5074 inhibit *P. falciparum* parasites in culture at low µM concentrations. *P. falciparum* cultures (0.25% parasitemia) were treated with increasing concentrations of the indicated compounds for 72 h, after which the HRP2-based sandwich ELISA was used to measure the HRP2 levels, determined by optical density. The results shown are from a single typical experiment performed in duplicate (SD shown), representative of a series of three independent experiments (see Table 5 for pooled data).

**Table 1 ijms-23-13899-t001:** Summary of NLS binding studies using AlphaScreen technology.

Binding Interaction	Kd (nM) *
PfIMPα:T-ag-NLS-GFP	2.9 ± 1.6
PfIMPα:TGS1-NLS-GFP	6.6 ± 0.3
PfIMPα:GCN5-NLS-GFP	2.5 ± 1.7
TgIMPα:T-ag-NLS-GFP	3.9 ± 1.1
TgIMPα:GCN5-NLS-GFP	5.0 ± 1.5
TgIMPα:TGS1-NLS-GFP	1.9 ± 0.7
MmIMPα/β1:T-ag-NLS-GFP	2.8 ± 0.2
MmIMPα/β1:TGS1-NLS-GFP	2.5 ± 0.1
MmIMPα/β1:GCN5-NLS-GFP	3.2 ± 0.2

* Results represent the mean ± SD (n = 2) for Kd values measured as per Figure 1.

**Table 2 ijms-23-13899-t002:** Summary of binding analysis data of IMPαs with MmIMPβ1 from AlphaScreen analysis.

Binding Interaction	Kd (nM) *
PfIMPα:MmIMPβ1	1.3 ± 0.3
TgIMPα:MmIMPβ1	1.2 ± 0.4
MmIMPα:MmIMPβ1	2.5 ± 0.2

* Results represent the mean ± SEM (n = 3) for Kd values measured as per Figure 2.

**Table 3 ijms-23-13899-t003:** Summary of data analysis for inhibition of IMPα-NLS-binding by ivermectin and GW5074.

	IC_50_ (μM) *	
Binding Interaction	Ivermectin	GW5074
PfIMPα:TGS1	5.0 ± 1.3	6.5 ± 0.4
TgIMPα:SV40 T-ag	4.9 ± 0.2	7.7 ± 1.9
ΔIBBmIMPα:SV40 T-ag	6.2 ± 0.2	4.9 ± 1.4

* Results represent the mean ± SEM (n = 3) for IC_50_ values measured as per Figure 3.

**Table 4 ijms-23-13899-t004:** Summary of data analysis for inhibition binding of IMPα to IMPβ1 by ivermectin and GW5074.

	IC_50_ (µM) *	
Binding Interaction	Ivermectin	GW5074
PfIMPα:MmIMPβ1	1.9 ± 0.2	41 ± 1.5
TgIMPα:MmIMPβ1	6.9 ± 1.2	11 ± 1.5
MmIMPα:MmIMPβ1	3.1 ± 0.5	22 ± 0.6

* Results represent the mean ± SEM (n = 3) for IC_50_ values measured as per Figure 4.

**Table 5 ijms-23-13899-t005:** Summary of IC_50_ values for inhibition of growth of *P. falciparum* parasites in culture.

Compound	IC_50_ (µM) *
Ivermectin	0.7 ± 0.1
GW5074	5.5 ± 0.9
DHA	0.001 ± 0.0005

* Results represent the mean ± SD (n = 3) for IC_50_ analysis as per Figure 6.

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
