# Peer review of "Conservation of Importin α Function in Apicomplexans: Ivermectin and GW5074 Target Plasmodium falciparum Importin α and Inhibit Parasite Growth in Culture"

_ijms, 2022, doi:10.3390/ijms232213899_

Round 1
Reviewer 1 Report
In this manuscritpt Sujata Walunj and collegues address the interesting issue of conservation of nucleocytoplasmic transport pathwasys in different organisms. SPecifically they compare mammalian and apicomplexa Importin alphas. They show that apicomplexa IMP alphas can work together with mammailian IMPbeta and recongize a number of common cargoes. They further show that inhibtors of mammalian IMPalpha such as IVM and GW5074 can inhibit apicomplexa replication at low micromolar concentrations, and affect apicomplexa IMPa conformation.
The manuscript, is well written and orgnized and the experiments carefully performed, contorlled and interpreted. The main limitations of the sudy are two and both related to the sentence
" The likely mechanism of action of iver-mectin in this context is through binding to PfIMPa, thereby disrupting its function in mediating nuclear transport”.
such sentence is neither corroborated by experimental evidence nor by the published literature.
1) Indeed the in vitro data on IMPa function is compelling but no in cells data have been shown. Immunofluorescence experiments of Apicomplexa cultured in the presence and in the absence of the inhibitors tested in this study need to be performed, aimed at investigating the subcellular localization of different nuclear proteins bearing cNLSs.
2) Iivermectin, also known as "the wonder drug", has been known for a long time for its broad antimicrobial actitivy. Its mehcanism of action is extremely pleiotropic, rnaging from binding to glutamate-gated chloride channels, to inhibiton of viral helicases. Therefore, ivermecting is acting on a number of cellular processes, which could well be responsible for the antimicrobial activity reported here. Hence, such possibilities need to be thoroughly discussed in the manuscript.
Author Response
We thank the Reviewer for the constructive comments and suggestions.
The manuscript, is well written and orgnized and the experiments carefully performed, contorlled and interpreted. The main limitations of the sudy are two and both related to the sentence
" The likely mechanism of action of iver-mectin in this context is through binding to PfIMPa,thereby disrupting its function in mediating nuclear transport”
such sentence is neither corroborated by experimental evidence nor by the published literature.
As we are sure the Reviewer understands, this is the first study to examine PfIMPalpha in any detail, and hence the published literature cannot address this point.
1) Indeed the in vitro data on IMPa function is compelling but no in cells data have been shown. Immunofluorescence experiments of Apicomplexa cultured in the presence and in the absence of the inhibitors tested in this study need to be performed, aimed at investigating the subcellular localization of different nuclear proteins bearing cNLSs.
We agree that our data is compelling in demonstrating direct binding to PfIMPalpha, but of course recognise that in vivo data is important to be able to confirm that this occurs in cells. As the Reviewer is likely aware, experiments in malarial parasites to determine subcellular localisation are a considerable undertaking because of the lack of robust molecular genetic tools/low efficiency of transfection etc., making such experiments well outside the scope of the present study. However, we have included the Reviewer's point in the discussion (line 340-343, and thank the Reviewer for this). We have also modified the “offending” statement (above) from the Discussion (line 332/333) to ensure it does not overstate our results in this context.
2) Iivermectin, also known as "the wonder drug", has been known for a long time for its broad antimicrobial actitivy. Its mehcanism of action is extremely pleiotropic, rnaging from binding to glutamate-gated chloride channels, to inhibiton of viral helicases. Therefore, ivermecting is acting on a number of cellular processes, which could well be responsible for the antimicrobial activity reported here. Hence, such possibilities need to be thoroughly discussed in the manuscript.
We agree with the Reviewer, and have now added these considerations to the Discussion (line 336-340) – again, we thank the Reviewer. The fact that both ivermectin and GW5074, with very different structures and other activities both target PfIMPalpha through direct binding, and both have antimalarial action implies that PfIMPalpha is the target in this case of both agents (line 334-336), but of course this needs to be demonstrated experimentally. We have incorporated this into the Discussion (as above - line 340-343) and thank the Reviewer again for encouraging us to include this in the Discussion.
Reviewer 2 Report
The present manuscript provides an interesting story. The manuscript is well conducted, with clear results and documented discussion and conclusion. It is well written in general, with some issues which have to be clarified/added/changed prior to the final decision. Detailed info can be found in the attached pdf file.

Author Response
We are very grateful to the Reviewer for the detailed “mark up” of our manuscript. Throughout, we have incorporated the recommended changes as far as possible, including opting for British spelling, defining abbreviations where lacking (eg. in the abstract), simplifying sentences where required and improving consistency.
After confirming this point with the editor, we have opted to persist with our format of Figure caption title/use of abbreviations, which is consistent with IJMS format guidelines based on our recent experience in publishing with IJMS – we prefer the Figure title to express the result depicted in the figure, helping understanding of the data and its interpretation. Since the abbreviations have all been defined clearly earlier in the manuscript, reducing the clutter in the Figure title by not redefining everything enables a shorter title that is concise and clear. We are sure the Reviewer can appreciate this point of view.
We thank the Reviewer again for the valuable input.Round 2
Reviewer 1 Report
The authors thoroughly modified the manuscript to address all the points raised. I feel the manuscript is ready for pubblication.